# Successful Introduction of Benralizumab for Eosinophilic Ascites

**DOI:** 10.3390/biomedicines12010117

**Published:** 2024-01-06

**Authors:** Nabil Belfeki, Nouha Ghriss, Souheil Zayet, Faten El Hedhili, Cyrus Moini, Guillaume Lefevre

**Affiliations:** 1Department of Internal Medicine and Clinical Immunology, Groupe Hospitalier Sud Ile de France, 77000 Melun, France; nouha.ghriss@ghsif.fr; 2Infectious Disease Department, Nord Franche-Comté Hospital, 90400 Trevenans, France; souheil.zayet@hnfc.fr; 3Department of Diagnostic Imaging, Groupe Hospitalier Sud Ile de France, 77000 Melun, France; faten.el-hedhili@ghsif.fr; 4Department of Cardiology, Groupe Hospitalier Sud Ile de France, 77000 Melun, France; cyrus.moini@ghsif.fr; 5Department of Internal Medicine and Clinical Immunology, University Hospital of Lille, 59037 Lille, France; guillaume.lefevre@chru-lille.fr; 6National Reference Center for Hypereosinophilic Syndromes, University Hospital of Lille, 59037 Lille, France

**Keywords:** benralizumab, eosinophilic ascites, eosinophilic gastroenteritis, steroids

## Abstract

Eosinophilic ascites is a rare disorder, reported in both adult and pediatric patients, characterized by high eosinophil counts in the peritoneal fluid. Eosinophilic ascites appears as a manifestation of various diseases such as parasitic and fungal infections, malignancy, and hypereosinophilic syndrome. It also represents an uncommon manifestation of eosinophilic gastroenteritis, usually treated with corticosteroids. We present the case of a 16-year-old woman with abdominal distention related to abundant ascites. Further work-up concluded that it was eosinophilic gastroenteritis complicated with eosinophilic ascites. The patient was on oral steroids for three weeks, but various abdominal relapses were observed, leading to the introduction of benralizumab, as a steroid-sparing therapy with a favorable outcome.

## 1. Introduction

Ascites often appears as a manifestation of various diseases such as parenchymal liver disease, malignancy, cardiovascular disease, or tuberculosis. Eosinophilic ascites is a rare condition. A high level of eosinophils in the peritoneal fluid is seen in parasitic, vasculitis, or malignant diseases [1]. Eosinophilic ascites (EA) is an unusual presentation of eosinophilic gastroenteritis (EGE) [2]. It causes a large spectrum of symptoms such as abdominal pain, diarrhea, nausea, vomiting, bloating, or ascites [3]. 

Clinical presentation depends on the predominant involved layer of the gastrointestinal tract according to the arbitrary Klein classification which divides patients with EGE into those with predominantly mucosal, muscle layer, or subserosal disease [4].

The overall prognosis for eosinophilic ascites is good, with an excellent response to oral steroids as a first-line treatment [5]. Some patients showed clinical relapse despite the corticosteroids. Little information is available regarding corticosteroids-sparing agents, but novel approaches including leukotriene receptor antagonists and monoclonal antibodies against IgE and IL-5 are mentioned. Benralizumab, a humanized monoclonal antibody against interleukin-5 receptor α, appears as steroid-sparing therapy for EGE [6]. Here, we report a case of EGE that presented with relapsing eosinophilic ascites (EG) which was successfully managed with benralizumab

## 2. Case Report

A 16-year-old female without personal or familial medical history presented to the emergency department with a febrile diffuse abdominal pain, diarrhea, nonbilious, and non-bloody vomiting. She denied any recent drug intake or recent travel. A physical examination showed a temperature of 38.5 °C, a heart rate of 89 beats per min, and a blood pressure of 140/80 mmHg. An abdominal examination showed epigastric tenderness, distention, and shifting dullness suggestive of moderate ascites. A cell blood count showed elevated leukocytes (19.78 G/L; normal range 4.0–10.0) with elevated eosinophil counts (13.0 G/L; normal range <0.5), hemoglobin at 12.3 g/dl (normal range 11.5–15 g/dl), and platelet count at 436 G/L (normal range 150–450 G/L). The C-reactive protein level was at 10.3 mg/L (normal range: <3 mg/L). Serum levels of liver enzymes, creatinine, and electrolytes were normal as well as serum levels of vitamin B12, folate, and ferritin. An abdominal and pelvic computed tomography scan confirmed ascites with gastric and duodenal wall thickening and submucosal edema (Figure 1).

An abdominal paracentesis showed low serum ascites albumin gradient (23.3 g/L) with elevated eosinophil count (leukocyte count 1000/mm^3^ with 890 eosinophils). Mycobacterial and microbiological cultures were negative, and no malignant cells were detected. Parasitological stool and urine examinations were negative as well as serological tests for parasitic infections including toxocariasis, bilharzia, strongyloidiasis, echinococcus, cysticercosis, trichinosis, distomatosis, and filariasis. Probabilistic anti-parasitic treatment with albendazole (400 mg bid for 14 days) and ivermectin (12 mg) was unsuccessful. Afterwards, an upper endoscopy revealed active duodenitis with significant eosinophilic infiltrate (50 eosinophils per high-power field. A colonoscopy showed colitis with mild inflammatory and intraepithelial eosinophils infiltrate throughout the whole colon region (≥20 eosinophils per high-power field) (Figure 2).

Immunological investigations including antinuclear antibodies and antineutrophil cytoplasmic antibodies were negative. Immunophenotyping of peripheral blood by flow cytometry did not show leukemia or lymphoma. Clonal populations of T- and B-cells in blood were negative. A bone marrow biopsy ruled out lymphoma and myeloproliferative neoplasms. Positron emission computed tomography did not reveal any neoplastic or infectious process. The diagnosis of idiopathic eosinophilic ascites revealing eosinophilic gastroenteritis was retained. Treatment consisted of oral prednisone at 30 mg daily for two weeks with progressive tapering until discontinuation at three months. The patient rapidly improved with resorption of ascites and normalization of blood eosinophilic count. But, six months later, while she was not on steroids, she relapsed with abdominal pain and enhancement of blood eosinophilic count (9 G/L; normal range <0.5). Three-month oral corticosteroids (30 mg/day of oral prednisone with progressive tapering) were renewed in association with leukotriene receptor antagonists. The patient improved but similar relapses occurred within the 3 months after discontinuing steroids. Thus, we considered benralizumab at the dosage of 30 mg every 4 weeks for 3 months followed by 30 mg every 8 weeks for a period of 18 months. During this period, we considered an effective oral contraception. We did not observe any relapse during a three-year follow-up and the cell blood count showed a total and persistent depletion of eosinophils. 

## 3. Discussion

Our patient had eosinophilic gastroenteritis with eosinophilic ascites, high peripheral blood eosinophilia, and an eosinophilic infiltrate of the duodenum. Eosinophilic ascites is defined by the presence of >100 eosinophils/µL or eosinophils comprising >10% of the non-erythrocyte count of the ascitic fluid [7]. It can be related to parasitic infestations (Strongyloides Stercoralis, Toxocara Canis), complicated hydatic cyst, bacterial peritonitis, inflammatory bowel diseases, solid tumors (ovarian cancer, peritoneal carcinomatosis), lymphoproliferative syndromes, and idiopathic hypereosinophilic syndrome. Physicians must be aware of the exhaustive list of conditions responsible for EA and thus a systematic approach is mandatory for establishing an etiological diagnosis. However, in some cases, eosinophilic ascites remains isolated as a single organ disease but further work-up including histological and laboratory investigations has led to the diagnosis of eosinophilic gastroenteritis in 2/3 of cases [1]. 

Eosinophilic gastroenteritis is a disease, first described by Kaijser in 1937, frequently affecting children as well as adults between 30 and 50 years old with a female predominance [8]. It is characterized by recurrent prominent eosinophilic infiltration of the digestive tract, and it presents with nonspecific gastrointestinal symptoms in association with peripheral eosinophilia. The etiopathogenesis of EG has not been elucidated. Previous reports highlighted a strong correlation with atopy. Around 80% of patients had a personal history of asthma, eczema, allergic rhinitis, or allergy. The eosinophil seems to be a major effector cell in generating inflammation in EGE with T-helper-2 type cytokines and eotaxin upregulation. EGE is defined by the presence of eosinophilic infiltration of the gastrointestinal tract or ascites with a predominance of eosinophils.

Clinical presentation seems to depend on the depth of eosinophilic tissue infiltration [9]. Klein et al. classified eosinophilic gastroenteritis into three pathologically types: predominant mucosal layer, predominant muscle layer, and predominant subserosal layer [4]. A predominantly serosal pattern is the rarest presentation of EG, in which eosinophil-rich inflammatory infiltrate permeates all layers of the digestive wall and reaches the serosal cover [10]. The principal manifestation in patients with subserosal disease is ascites with a high eosinophilic count in the peritoneal fluid [4]. NJ Talley showed that subserosal involvement was associated with a higher peripheral eosinophilic count and had the best response to steroids [11]. Table 1 summarize the reported cases of eosinophilic ascites with clinical presentation, management, and outcome.

Treatment with steroids ended with a good response in our patient, but clinical relapses were observed. There is no consensus regarding treatment, but the main goal is to reduce symptoms and eosinophil infiltration as well as eosinophil count. Corticosteroids usually induce remission within a two-week period. An appropriate duration of steroid treatment is unknown and relapse often necessitates long-term treatment. Unfortunately, long-term steroid treatment predisposes some patients to serious side effects [9]. Some authors propose steroid-sparing therapy strategies such anti-histamines, mast cell inhibitors, leukotriene receptor antagonists, and anti-interleukin drugs including ketotifen in the treatment of relapses to avoid the side-effects of steroids [12]. In our case, benralizumab was considered as clinical relapse was observed despite the corticosteroids and leukotriene receptor antagonists. Interleukin-5 (IL-5) is essential to the eosinophil life cycle, stimulating the proliferation, differentiation, and maturation of eosinophil precursors. IL-5 is a crucial cytokine implicated in the pathogenesis of eosinophilic gastroenteritis [13,14]. Benralizumab is a humanized monoclonal antibody against the interleukin-5 receptor α (IL5RA) that depletes eosinophils. Benralizumab binds to the IL-5 receptor alpha subunit, preventing eosinophil signal transduction-inducing antibody-dependent cell-mediated cytotoxicity by natural killer (NK) cells. In November 2017, the FDA approved the drug for use as add-on maintenance therapy for patients with severe eosinophilic asthma with an improvement in lung function and a reduction of exacerbations [15,16]. In a small phase 2 trial, a placebo-controlled phase then an open-label phase, including patients with hypereosinophilic syndrome with persistent disease (an absolute eosinophil count of at least 1000 cells per cubic millimeter while receiving stable therapy for at least 1 month) received benralizumab at a dose of 30 mg every 4 weeks. A reduction of at least 50% in the absolute eosinophil count at week 12 occurred in more patients in the benralizumab group than in the placebo group (*p* = 0.02). At week 48, the percentage of patients who had a hematologic and clinical response to the benralizumab therapy was 74% [16]. Miguel L Stein’s study that included four patients with eosinophilic esophagitis demonstrated that anti–IL-5 therapy was associated with improvements in clinical symptoms, quality of life, endoscopic findings, peripheral blood eosinophilia, and pathological features of eosinophilic esophagitis [17].

## 4. Conclusions

Eosinophilic gastrointestinal diseases are a group of heterogeneous diseases characterized by eosinophilic infiltration of the gastrointestinal tract, and they can present as a variety of gastrointestinal symptoms. Eosinophilic ascites is the most unusual presentation of eosinophilic gastroenteritis caused by the eosinophilic inflammation of the small bowel wall’s serosal layer. It should be considered in the absence of malignancy, infectious disease, or hematologic malignancy. Steroids are the first-line treatment but relapses are frequent. This case illustrates that benralizumab, humanized monoclonal antibody against the interleukin-5 receptor, represents an innovative alternative therapy for eosinophilic gastroenteritis. Further larger studies are needed to determine the place of benralizumab in the management of an eosinophilic single-organ disease such as EGE and in the duration of the treatment after sustained remission induction.

## Figures and Tables

**Figure 1 biomedicines-12-00117-f001:**
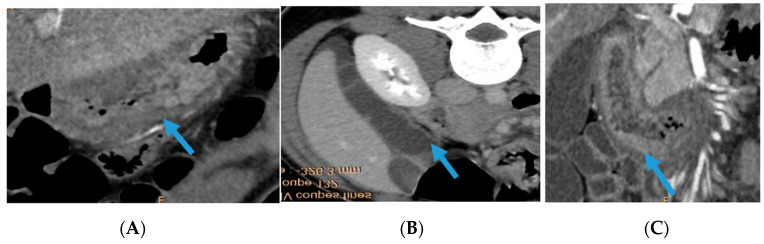
Abdominal and pelvic computed tomography; (**A**) coronal section of an abdominal CT scan with contrast product injection showing gastric wall thickening and submucosal edema (blue arrow); (**B**) axial section of an abdominal CT scan with contrast product injection showing ascites (blue arrow); (**C**) coronal section of an abdominal CT scan with contrast product injection as well as duodenal wall thickening and submucosal edema (blue star).

**Figure 2 biomedicines-12-00117-f002:**
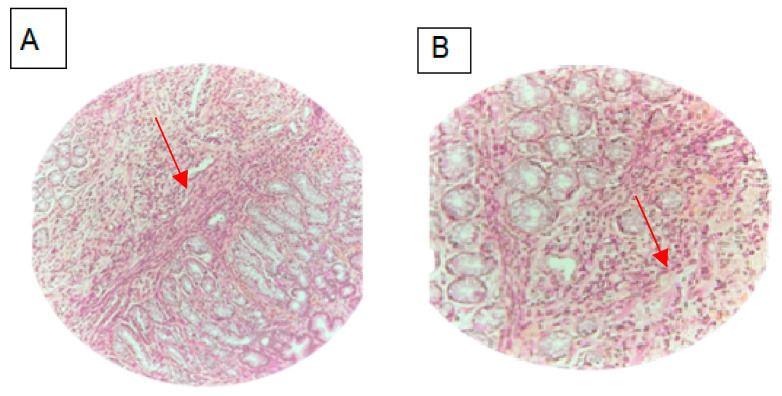
Biopsies showed mucosa with interstitial edema and eosinophilic infiltration throughout the duodenum (hematoxylin and eosin, (**A**) magnification 20×; (**B**) magnification 40×).

**Table 1 biomedicines-12-00117-t001:** Review of the literature of reported eosinophilic ascites related to eosinophilic gastroenteritis.

	Cuko, L [18] 2014	Agrawal, Shefali [12] 2016	Ferreira, António- Araújo [19] 2017	Martín-Lagos Maldonado, Alicia [5] 2018	Santos, Carina [20] 2018	Feng, Wan [21] 2020	Tian, Xiao-Qing [22] 2021	El Ray, Ahmed [23] 2021	Our Case
Gender	Woman	Woman	Woman	Man	Woman	Man	Man	Woman	Woman
Age	37	34	41	35	32	26	34	30	16
Abdominal pain	+	+	+	+	+	+	+	+	+
Diarrhea	+	+	+	-	+	+	+	-	+
Abdominal distension	-	-	+	-	+	+	-	+	+
Nausea	+	-	-	-	+	-	-	+	+
Blood eosinophil count	4.94 G/L	11.9 G/L	3.454 G/L	2.969 G/L	4.88 G/L	8.2 G/L	4.85 G/L	9.45 G/L	13.0 G/L
Intestinal wall thickening	+	-	+	+	+	+	+	+	+
Ascitic fluid analysis	94% Eo	100% Eo	Inflammatory liquid with predominance of Eo	95% Eo	93.3% Eo	High eosinophil count	-	90% Eo	89% Eo
Endoscopic findings	Normal	Eosinophilic gastritis and duodenitis	Normal	Normal	Eosinophilic colitis	Eosinophilic gastritis and colitis	Eosinophilic colitis	Normal	Eosinophilic duodenitis and colitis
Treatment	Prednisone (40 mg/d than 15 mg/d)	Prednisone (25 mg/d)	Prednisone (40 mg/d)	Prednisone (25 mg/d)	Prednisone (40 mg/d)	Prednisone (40 mg/d)	Prednisone (40 mg/d) 6 months	Prednisone (40 mg/d) 1 month	Prednisone (30 mg/d) 3 months
Outcome	improvement	improvement	relapse	improvement	improvement	improvement	improvement	improvement	Relapse
Follow-up	ND	2 years	ND	2 months	ND	ND	1 year	3 months	4 years

ND: not determined; Eo: eosinophils.

## Data Availability

The data presented in this study are available on request from the corresponding author.

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
