# Peer review of "Successful Introduction of Benralizumab for Eosinophilic Ascites"

_biomedicines, 2024, doi:10.3390/biomedicines12010117_

Round 1
Reviewer 1 Report
Comments and Suggestions for Authors
The paper, titled as “Successful Benralizumab in eosinophilic ascites”, by Nabil Belfeki, to explore the therapy of Benralizumab in eosinophilic ascites. This was an interesting study. However, there are something that needed to clarify before drawing some conclusions.
1. The case report is very clear, but it is best to supplement post-treatment data, such as CT images after treatment.
Comments on the Quality of English LanguageGrammar can be moderately improved.
Author Response
Dear Sir or Madam,
Thank you very much for your comments.
We added details concerning laboratory data during relapses. Moreover, we added the duration of biotherapy and mentionned that the patient was on effective oral contraception.
Reviewer 2 Report
Comments and Suggestions for Authors
The authors described an interesting case of eosinophilic gastroenteritis successful treated with benralizumab. However, I have several questions.
1. This case report is mainly focused on eosinophilic gastroenteritis, while eosinophilic ascites is one of its manifestations. I suggest using eosinophilic gastroenteritis instead.
2. Have you done a biopsy of the peritoneum ?
3. The patient was presented with disease flare-remitting pattern. What was the dosage of prednisone at the first relapse? A detailed description of flare manifestations should be added, such as the eosinophilic count and quantification of the ascites by US/CT.
4. Did the patient receive benralizumab at the dose of 30mg every 4 weeks for all the 3 years? or was there a taper ?
5. How was the prednisone tapered with the help of benralizumab ? Was the prednisone remained free for all the 3 years?
6. A time diagram including factors such as prednisone dosage, eosinophilic count, quantifications of ascites and the administrations of benralizumab would better illustrate the flare pattern and the efficacy of benralizumab.
Author Response
Dear Sir or Madam,
Thank you for accepting reviewing our draft.
The aim of our work was to add eosinophilic gastro enteritis as a causal condition of eosinophilic ascites. That's why the first part of the clinical case was to determine how we should approach eosinophilic ascites and rule out other possible related disease. The second concern of our draft was the management of this relapsing disease. That's why, we prefer to talk about Eo ascites.
We did not perform a peritoneal biopsy. We perform only a paracentesis with cytological analysis to rule out solid tumor or lymphoma.
Each time the patient relapsed, she was not on steroids. We added this informations in the text. Clinical manifestations consisted on abdominal pain only and we added biological data at relaspe.
We mentionned that biologics were only for 18 months. Modifications were made in the text.
There is not enough data in our medical record to perform a synthesis as recommended. moreover, the relapse concern only a reccurent abdominal pain with an enhancement of eosinophil count. We apologize.
Reviewer 3 Report
Comments and Suggestions for Authors
Reviewer:
Peer review of the document entitled: “Successful Benralizumab in eosinophilic ascites” by Nabil Belfeki1 and coworkers.
General comments:
The aim of the present single case report was to evaluate the role of benralizumab in a young patient afflicted with eosinophilic ascites. The manuscript lays within the Journal's scope.
Specific Comments:
Introduction:
Line 33: The estimated prevalence for eosinophilic gastroenteritis (if available) should be indicated and referenced.
Line 44: These evidences should be referenced.
Case report:
Line 70: The absolute number of eosinophils should be also included here.
Line 98: Did the patient, and/or her parents/guardians signed a specific informed consent form before undergoing therapy with benralizumab as an off-label use? This is of special interest for those <18 y.o. Please confirm and clarify.
The currently approved dosage of benralizumab 30 mg for severe uncontrolled eosinophilic asthma is QW8. Why was benralizumab 30 mg given QW4 instead of QW8? Please, address this point as hypereosinophilic syndrome is a much rarer condition than uncontrolled severe eosinophilic asthma, and in fact only mepolizumab 300 mg QW4 is the only currently approved biologic therapy for HES.
Also indicate the current peripheral whole blood eosinophils count after treatment (and monitoring) with benralizumab, as is generally depleted to 0.0.
Discussion:
Line 144: The following text: “It has been demonstrated that Benralizumab reduced exacerbations and improved lung function in patients with severe eosinophilic asthma. In November 2017, the FDA approved the drug for use as add-on maintenance therapy for patients with severe eosinophilic asthma [15,16].” should be merged into a single sentence to add clarity.
Line 156: Limitations (or yet unsolved issues) to the present case report should be also mentioned, as:
-Could the currently benralizumab dosage be modified after 3 years of treatment?
-How long should be benralizumab continued (still off-label use) if the patient remains clinically stable after 3 years?
Comments on the Quality of English Language
N/A
Author Response
Dear Sir or Madam,
Many thanks for accepting reviewing our draft.
In the published data, there is no real prevalence of Eo ascites related to Eo GE. Nevertheless, we tried to epidemiological data of Eo ascites in a table.
We modified the draft and specified the absolut count of eosinophils.
We confirm that the parents signed a specific consent before undergoing therapy with benralizumab
We clarified in the text our therapeutic approach. we added benralizumab at the dosage of 30 mg every 4 weeks for three months than 30 mg every 8 weeks. The treatment was maintained 18 months.
Eosinophils were completely depleted after starting benralizumab. This detail was added in the text.
As mentionned by reviwers, we added at the end a phrase showing the limitation of this case. In fact, further larger studies are needed to determine the place of benralizumab in EGE and the duration of this biotherapy after induction of remission.
Round 2
Reviewer 2 Report
Comments and Suggestions for Authors
Thanks for the revision. I have no further comments.